# Investigation of Plasma-Derived Lipidome Profiles in Experimental Cerebral Malaria in a Mouse Model Study

**DOI:** 10.3390/ijms24010501

**Published:** 2022-12-28

**Authors:** Amani M. Batarseh, Fatemeh Vafaee, Elham Hosseini-Beheshti, Azadeh Safarchi, Alex Chen, Amy Cohen, Annette Juillard, Nicholas Henry Hunt, Michael Mariani, Todd Mitchell, Georges Emile Raymond Grau

**Affiliations:** 1Sydney Knowledge Hub, BCAL Dx Ltd., The University of Sydney, Merewether Building, Sydney, NSW 2006, Australia; 2BCAL Dx Ltd., Suite 506, 50 Clarence St., Sydney, NSW 2000, Australia; 3School of Biotechnology and Biomolecular Sciences, University of New South Wales, Sydney, NSW 2052, Australia; 4UNSW Data Science Hub, University of New South Wales, Sydney, NSW 2052, Australia; 5OmniOmics.ai Pty Ltd., Sydney, NSW 2035, Australia; 6Vascular Immunology Unit, Discipline of Pathology, School of Medical Sciences, Faculty of Medicine & Health, The University of Sydney, Sydney, NSW 2000, Australia; 7Thermo Fisher Scientific, Scoresby, VIC 3179, Australia; 8Faculty of Medicine & Health, Bosch Institute, School of Medical Sciences, The University of Sydney, Sydney, NSW 2000, Australia; 9Thermo Fisher Scientific, North Ryde, NSW 2113, Australia; 10School of Medicine, Faculty of Science Medicine and Health, University of Wollongong, Wollongong, NSW 2522, Australia; 11Illawarra Health and Medical Research Institute, University of Wollongong, Wollongong, NSW 2522, Australia

**Keywords:** cerebral malaria, lipidome, mouse model, *Plasmodium* spp.

## Abstract

Cerebral malaria (CM), a fatal complication of Plasmodium infection that affects children, especially under the age of five, in sub-Saharan Africa and adults in South-East Asia, results from incompletely understood pathogenetic mechanisms. Increased release of circulating miRNA, proteins, lipids and extracellular vesicles has been found in CM patients and experimental mouse models. We compared lipid profiles derived from the plasma of CBA mice infected with *Plasmodium berghei* ANKA (PbA), which causes CM, to those from *Plasmodium yoelii* (Py), which does not. We previously showed that platelet-free plasma (18k fractions enriched from plasma) contains a high number of extracellular vesicles (EVs). Here, we found that this fraction produced at the time of CM differed dramatically from those of non-CM mice, despite identical levels of parasitaemia. Using high-resolution liquid chromatography–mass spectrometry (LCMS), we identified over 300 lipid species within 12 lipid classes. We identified 45 and 75 lipid species, mostly including glycerolipids and phospholipids, with significantly altered concentrations in PbA-infected mice compared to Py-infected and uninfected mice, respectively. Total lysophosphatidylethanolamine (LPE) levels were significantly lower in PbA infection compared to Py infection and controls. These results suggest that experimental CM could be characterised by specific changes in the lipid composition of the 18k fraction containing circulating EVs and can be considered an appropriate model to study the role of lipids in the pathophysiology of CM.

## 1. Introduction

Malaria is one of the significant major concerns in global public health, especially in endemic areas, with over 241 million cases in 2021 in more than 85 countries as reported by the World Health Organization (WHO, 2021 Malaria report, available from 6 December 2021, https://www.who.int/publications/i/item/9789240040496). Severe malaria is a multi-syndromic infection caused by *Plasmodium falciparum* and often manifests as cerebral malaria (CM), severe malaria anaemia, acute renal failure and respiratory failure [1,2]. CM is characterised by unarousable coma (Glasgow coma scale < 11, Blantyre coma scale < 3), neurological deficits and neurological sequelae [2,3]. This debilitating syndrome accounts for the majority of malaria-induced deaths annually [4,5,6] (2021 report, WHO).

The pathogenetic mechanisms of CM are exceedingly complex and, therefore, incompletely understood. Dynamic interactions between infected erythrocyte sequestration, host cell activation and inappropriate immuno-inflammatory responses have been extensively studied [1,7,8,9,10,11,12]. Due to the complexity and difficulty of performing experimental investigations in humans, mouse models of CM have been used to better understand CM pathogenicity. Despite the fact that the murine CM model has limits (e.g., including differences in pathophysiology, parasite sequestration in the brain, cytopathological evidence of inflammation and microvascular sequestration of parasitised erythrocytes compared to human CM [13,14], the murine model has several positive aspects. Both human CM and experimental CM in a mouse model can be characterised by severe vasculopathy, upregulation of inflammatory cytokines, microhaemorrhages leading to neurological impairment and the concordance between immune regulatory mechanisms [14,15,16,17,18,19]. A widely used model for CM is inbred CBA mice infected with *Plasmodium berghei* ANKA (PbA), which leads to fatal disease with neurological signs such as paralysis, ataxia, convulsion and/or coma and finally death within ten days [20,21]. Conversely, infection of CM-susceptible mice with *Plasmodium yoelii* (Py) leads to hyper-parasitaemia and anaemia but without neurological complications [22]. This syndrome following Py infection is referred to as non-cerebral malaria (NCM).

Lipid subspecies/families have emerged as important regulators of pathophysiological conditions in vitro [23] and in vivo [24]. The roles of lipids and fatty acids are increasingly known in inflammation, immunoregulation, metabolism and cancer [25], as well as in malaria parasite biology [26,27]. A reduction in lysophosphatidylcholines (LPCs) and monounsaturated fatty acid-containing phospholipids and an elevation in fatty acylcarnitines was reported in malaria that may be used by the parasite to build its own membrane during the acute stage of infection [28,29]. Lipids also play a key role in the biology of extracellular vesicles (EV), which are key players in cell–cell interactions as well as CM pathogenesis [12,30,31]. Microvesicles (MV), previously called microparticles (MP), are one of the four families of EV and now recognised as major elements in cell–cell communications [32], notably in the central nervous system [33]. They play essential roles in homeostasis and are active players in inflammatory and immunopathological conditions [34], including CM [31,35].

Previous studies showed that two-step centrifugation results in platelet-free plasma (PFP) with high concentrations of EVs [36]. We previously performed two-step centrifugation (1500× *g* for 15 min and 1800× *g* for four minutes (twice)) [12] that resulted in a high number of MPs in the mouse-collected plasma known as “18k pellet” or, according to ISEV recommendations, “large EV” with >100 nm diameter as opposed to “small EV”, which would be the exosome fraction with 30–100 nm diameter, and usually are isolated by 10,000× *g* centrifugation for 20 min [37,38] and are outside of the scope of the current study. The aims of this study were to determine whether the profiles of PFP-derived lipids during CM and NCM differed and to evaluate whether some lipid species could be correlated with pathogenesis and might be used as biomarkers of disease severity and/or targets for therapeutic intervention.

## 2. Results

### 2.1. Qualitative and Quantitative Changes in EV/MV Produced in CM versus NCM

Plasma was purified, and PFP-derived lipids were extracted, as described, from the three groups of control, PbA-infected (i.e., with CM) and Py-infected (i.e., NCM) mice. Compared to those from controls, MV from PbA-infected mice showed a doubling (29.43% vs. 15.28%) of their proportion of triglycerides (TG), a 25% reduction (47.6 vs. 62.6%) in their cholesteryl ester (CE) proportion and a 50% reduction (2.8 vs. 5.6%) in their lysophosphatidylcholine (LPC) content (Figure 1A). These MV also presented a threefold increase (2.2 vs. 0.7%) in their diacylglycerol (DG) content. Conversely, MV from Py-infected mice did not show such differences when compared to those from uninfected control mice. Parasitaemia levels were not significantly different between PbA- and Py-infected animals (not shown) [39].

There have been reports suggesting that cholesteryl ester and TG are contaminants in MV preparations [23]. The levels of TG and CE in the preparations are higher than other lipid classes; therefore, we also analysed the results without these two classes of lipids (Figure 1B). Under these criteria, the reduction in LPC in MV from PbA-infected animals was confirmed, and a 3-fold increase in hexosylceramides (HexCer) was disclosed as well as a 5-fold increase for Py-infected mice.

### 2.2. CM Caused Significant Alterations in MV-Derived Lipid Class Composition

From 12 lipid classes, significant changes were observed in the levels of DG, LPC, lysophosphatidylethanolamine (LPE), phosphatidylcholine (PC), phosphatidylethanolamine (PE) and phosphatidylserine (PS) lipid classes in MV from PbA-infected vs. control mice (Figure 2A,B). A quantitative analysis of these differences identified that, of these classes, LPC and LPE were decreased, and the rest were increased significantly. On the other hand, in MV from Py-infected vs. control mice, DG, HexCer, ceramide (Cer), PE, LPC, PC and PS lipid class levels were significantly modified. When comparing MV from PbA-infected to those from Py-infected mice, although the changes were observed in most of the lipid classes (Figure 2A), only LPE amount was significantly reduced in MV from PbA-infected mice (Figure 2B).

A principal component analysis (PCA) of lipidomes derived from MV (18k PFP) lipidomes in the three groups of mice illustrates that lipids in PbA were the most different from the ones in control conditions (Figure 3A). The numbers of differentially expressed lipids (i.e., adjusted *p*-value < 0.01 and |log2 fold-change| > 1) from the three categories of mice were further visualised using volcano plots and Venn diagrams (Figure 3B–D). The lipidome from the PbA group differed from the ones from the Py group, with both increased and decreased lipid species (Figure 3B,C). Volcano plots demonstrated that the lipidome of PbA-infected mouse was dramatically different compared to controls, while that of Py-infected mice was not (Figure 3B). When compared to controls, the abundance of 29 lipid species was increased in PbA-infected mice, and 52 lipid species decreased. In contrast, Py-infected mice showed only nine lipid species increased and two lipid species decreased. Interestingly, when compared to those from Py-infected MVs, PbA had 20 increased lipid species and 30 decreased lipid species (Figure 3C). The Venn diagram demonstrates that the lipidome of PbA-infected MVs was the most strikingly modulated (Figure 3D). It also shows substantial overlap among differentially modulated lipids in ‘PbA vs. Py-infected animals’ and ‘PbA-infected vs. uninfected control mice,’ which suggests similarity of the lipidomic profiles in Py-infected and uninfected mice when compared with PbA.

### 2.3. Quantitative Analysis of Lipid Species among Measured Lipid Classes

A comprehensive analysis of the lipid species was performed within each detected lipid class, and the composition of the lipid species was characterised. Table 1, Table 2 and Table 3 show the differentially expressed lipid ions in three groups of comparisons, PbA-infected vs. control mice (Table 1), Py-infected vs. control mice (Table 2) and PbA- vs. Py-infected mice (Table 3). Lipid species with a positive fold change are shown in blue, and negative fold change in red.

In PbA MV, all PE species were higher than control except PEO-16:1_18:2 and PEO-18:1_18:2 (Table 1). Compared to control mice, Py-MV had only three PE lipid species that were significantly increased (Table 2), in contrast to the twelve that were increased in PbA MV. When comparing PbA-infected to those from Py-infected mice, we found seven PE lipid species increased and two decreased (Table 3). No LPE species were significantly modulated in Py (Table 2). In contrast, LPE18:1 and LPE18:2 were reduced in both PbA vs. control (Table 1) and vs. Py (Table 3).

A striking number of identified LPC lipid species were significantly reduced in PbA vs. control and vs. Py, but there was no difference in the Py vs. control. Twelve PC species were significantly modulated in PbA vs. control (Table 1), and Py showed only two PC species in lower amounts than in controls (Table 2), despite no significant difference in total PC amounts (Figure 2). Interestingly, PC18:0_22:6 was higher in PbA compared to both control and Py. 

PS18:0_22:6 lipid amounts were higher in both PbA and Py vs. controls (Table 1 and Table 2), while two additional PS lipid species were significantly modulated in Py vs. control (Table 2). With regard to HexCer lipid species, both HexCer 18:1;O/16:0 and HexCer 42:1;O2 species were higher in PbA- and in Py-MV than in controls (Table 1 and Table 2). However, HexCer 41:1;O2 was significantly lower than control in PbA only (Table 1). Moreover, SM 36:0;O2 was significantly increased, and PI O-34:3 was significantly decreased compared to control only in PbA-MV (Table 1). In the DG class, six species were found to be significantly higher in PbA-MV than in controls (Table 1), and three were significantly higher in PbA than in Py (Table 3). No DG lipids were significantly modulated between Py and control (Table 2). No changes were observed in CE and Cer lipids in our study.

Finally, we detected TG lipids in the 18k PFP preparations, and we comprehensively characterised the fatty acyl chain composition of the lipid species. Despite the lack of difference between the three groups at the TG class level (Figure 2), numerous TG species were differentially expressed (Table 1 and Figure 4). Interestingly, the five TG species containing docosahexaenoic acid (DHA FA 22:6) in their composition were the most strikingly different: all of them were higher, while all other identified TGs were lower, in PbA MV than in both Py and control, as highlighted in yellow boxes (Figure 4). Remarkably, there was no difference in any identified TG species between Py and controls.

The correlation heatmap (Figure 5) shows pairwise correlations among 121 lipids retained after removing invariant lipids (interquartile range, IQR ≤ 1) across all the samples. Changes are either in concordance (positive correlation shown in blue) or inverse concordance (negative correlation shown in red). Hierarchical clustering of the complete lipid–lipid correlation matrix describing 7260 unique pairs of lipids (excluding self) revealed eight distinct clusters of lipids positively correlated (Pearson correlation > 0.7) across all samples, organised along the diagonal of the matrix. Clusters 1 and 3 are highly expressed in PbA vs. control or vs. Py and essentially are composed of TGs and DGs. In contrast, cluster 2 is composed of lipids that are increased in PbA and Py vs. control but not between PbA and Py (Figure 5A). All other clusters, excluding cluster 8, are lowly expressed in PbA vs. control or Py. Interestingly, cluster 8 is increased and composed mostly of HexCer, PE and PS and has a similar change in both PbA and Py vs. control. Cluster 3 shows TG and DG lipids that are composed of 22:6 in their structure and have positive fold change in PbA vs. control, which is potentially related to CM pathogenesis. Cluster 3 shows a negative correlation with cluster 4, which is mostly composed of phospholipids, which are lowly expressed in PbA vs. control and Py (Figure 5A). Clusters 6 and 7 are the largest, with cluster 6 composed mostly of TGs and cluster 7 composed mostly of PC and LPC, both with negative fold change in PbA vs. control and vs. Py and could play a role in CM phenotype progression that is opposite to that of TG containing 22:6 in cluster 3.

## 3. Discussion

Numerous studies have investigated the serum lipid profile changes during malaria infection and suggested lipids being used as a biomarker [29,40,41]. In addition, the elevated changes of EVs (extracellular vesicles) in patients suffering from malaria, especially the ones with neurological complications such as CM and its association with pathogenicity and severity of the disease, were previously reported [35,42,43]. It is revealed that MVs and exosomes may have a role in post-transcriptional regulation of gene expression, innate immune system activation and modulation of the malaria transmission stage [31]. The level and content of EVs, including protein and miRNA, have also been investigated in a few CM mouse model studies that may be useful for biomarker analysis [44,45]. Several protocols were previously described and validated to isolate EV from various sources, from cell culture to human plasma, and the main gold standard is differential centrifugation to eliminate cells and cellular debris, followed by ultracentrifugation [46]. We previously validated our method for EV purification [12] and used it for this study to make sure about the robustness of the findings.

Previously, 300 lipids in *P. falciparum* asexual blood stage and gametocytes were profiled in the in vitro study by Gulati et al. [47]. Here, we used *P. berghei* ANKA (PbA)- and *P. yoelii*-infected mice representing experimental CM and non-CM models, respectively. As for human CM, experimental CM with PbA causes neurological signs and histopathological changes in the brain, including hemiplegia, paraplegia, convulsions, ataxia, microhaemorrhages, oedema, etc. [19,48]. However, histological alterations were not observed in the brain of infected mice with *P. yoelii* [19,49].

Total LPC and PC levels were significantly lower in PbA- and Py-infected mice compared to uninfected mice. Total LPE levels were also lower in PbA compared to uninfected mice and Py-infected mice. It is the only lipid class that significantly differs in PbA- and Py-infected mice. Activation of phospholipase A 2 (PLA2) is required to cleave the fatty acid on the sn2 position in phospholipids; this most commonly releases arachidonic acid, which is then converted to inflammatory eicosanoids. The other product is lysophospholipids such as LPC and LPE. The significant decrease in LPC in our study is concordant with other studies in human malaria and mouse model [40,47]. Gualti et al. reported a 6-fold and 4-fold reduction in LPC during early and late gametocytogenesis of *P. falciparum*, respectively [47]. Lower concentrations of circulating LPC are associated with parasitaemia and higher consumption of lipids during gametocytogenesis of the parasite to build its own membrane. Pappa et al. showed a positive correlation between PLA2 activity and neural inflammation in infected children [50]. The lower level of LPC may thus prevent tissue repair in the brain and affected organs in CM mice. Based on current knowledge, the reduction in LPE and LPC in MVs in CM suggests a lack of inflammation. However, these differences observed in plasma MV between PbA and Py infection are consistent with the strong immunopathological response underpinning CM, as opposed to non-CM, as detailed in several works [7,31,51,52,53]. For instance, Nacer et al. reported no brain histological symptoms in Py-infected mice [19]. PLA2 activation and increased level of LPC can enhance the function of immune cells, and can play an important regulatory role in both natural and adaptive immunity by inducing the expression of IFN-γ, heparin-binding epidermal growth factor-like growth factor (HB-EGF) and IL-2 receptor on human T lymphocytes. For further information about the role of LPE and LPC on immune cells, interested readers can refer to reviews by Lie et al. [54] and Kabarowski [55]. Investigating PLA2 levels and activity in CM MVs is of interest to elucidate if it plays a role in modulating the levels of MV lysophospholipids and, consequently, the development of CM. 

We also found that PE and PS levels were significantly higher in both PbA and Py compared to uninfected control, suggesting their possible role in malarial infection, but not CM pathogenesis specifically. This conclusion, however, is subject to further investigation. The increased level of PS in RBC-derived microvesicles (RMVs) was also reported in other studies from *P. falciparum* cultures or plasma of *P. falciparum*-infected humans or *P. berghei*-infected mice [43,47,56].

A higher number of lipid species were changed in CM mice vs. non-CM mice, which may show the importance of lipidome profile changes in the pathophysiology of CM and needs further investigation. Our finding showed the amount level of 75 lipid species significantly changed in MV of PbA-infected mice compared to the control group with 25 increased and 50 decreased. However, in Py-infected mice, only 10 lipid species had significant changes, with only two decreased. Furthermore, 45 lipid species had significant changes in PbA compared to Py group. Of these, 17 had an elevated amount, while 32 dropped. TG (27 lipid species), followed by LPC (9 species), were the lipid classes with the most significant changes. It is observed that lipid peroxidation increased in CM, which may lead to changes in the lipid profile and immunity responses. The increased lipid peroxidation and accumulation of triglycerides were detected in the mice infected with *P. berghei* NK65 [57] and PbA that includes the release of TNF-α and IFN-g, which have a key role in inflammation in the host [27]. Furthermore, diets supplemented with SFA, EPA and DHA fortified with alfa-tocopherol show reduced parasitaemia and improved survival rates in mice infected with *P. yoelii* due to their inhibitory and neutrophil antimalarial activity [27]. Although the changes in the plasma lipids have not been thoroughly studied in CM and need further investigation, these changes in lipidome profile may be used as a biomarker for CM diagnosis or progression as some of the lipid species were previously shown to change in severe malaria and CM patients [27,29].

DHA 22:6 have a known role in inflammation as they belong to the omega-3 pathway [58] and thereby are anti-inflammatory. In our study, 13 out of 16 DHA (22:6)-containing lipids were found to be increased, while they were decreased in 2 out of 16 lipids in PbA vs. control, suggesting an anti-inflammatory potential at the time of CM. Although DHA plays a protective role in inflammation, deleterious effects have also been reported, where an excess of omega-3 membrane lipids can increase the susceptibility to infection [59].

It has been suggested that elevated levels of TG lipids in MV preparations are considered an indication of contamination from other vesicle types present in plasma [60]. Surprisingly, despite the total level of detected TGs in our preparations being comparable between the three types of MV populations, TG lipids containing DHA 22:6 were exclusively increased in MV preparations from PbA-infected mice, indicating the potential utility of these lipids to predict CM development in mice.

Given that MVs are taken up by macrophages, it is tempting to speculate that the changed lipidome of MVs during CM plays an anti-inflammatory role or is a mechanism used by the parasite to modulate the host immune response [53]. More specifically, it is possible to hypothesise that MVs utilise TGs to supply DHA 22:6 to recipient cells that incorporate it into membrane phospholipids. Whether the role of DHA is deleterious (by increasing membrane fluidity or affecting the MV’s ability to fuse with target cells) or protective (by anti-inflammatory properties) remains to be elucidated.

Correlation maps showed clusters of lipid species that changed together in a related fashion; for instance, the changes in TG and DG correlated, which is not surprising since they are biologically linked, sharing similar biosynthetic pathways, and their increase has been reported previously and suggested that these lipids may have a key role in parasite growth and development noting that they are not abundant in RBCs [47]. The possibility that TG is being used as a storage site for fatty acids that can later be hydrolysed and employed for the synthesis of additional lipids remains of interest for future investigation.

In summary, this study is a preliminary but thorough assessment to evaluate if the plasma-derived lipid profiles significantly differ in mice infected by two different strains of malaria and to evaluate if lipid profiles can be correlated with pathogenesis. We aimed to explore alterations at the lipid species level instead of the class level, as individual lipids can be interesting for directing new research efforts. DHA-containing lipids may indicate a need to explore omega-3 and omega-6 lipids and how they are being utilised in the different groups. It highlights that there are distinct differences in the plasma vesicle lipid profiles of mice infected with two different malaria strains. These results suggest that experimental CM is characterised by the changes in contents of MVs (18k PFP) and, more specifically, by the reduction in LPC and LPE and a specific increase in TGs. Microvesicles carry a large array of active molecules, including lipid mediators, phospholipases, proteins and RNA, that can be used to modulate the phenotype of recipient cells [61]. Future studies investigating differences in lipid packaging and phospholipase activity in microvesicles from CM vs. NCM may shed some light on the role these lipids play in malaria complications.

## 4. Materials and Methods

### 4.1. Mice and Parasite Inoculation

We confirm that all experiments were performed in accordance with relevant guidelines and regulations. All mice used in this study were handled according to protocols approved by the University of Sydney Animal Ethics Committee (approval numbers K20/7-2006/3/4434, 418 and 326). Female CBA mice, 7 weeks old, were purchased from the Animal Resources Centre (Canning Vale, Western Australia). Mice were fed a commercial rodent pellet diet and had access to water ad libitum. Experimental mice were studied under pathogen-free conditions and monitored daily.

PbA was a personal gift from Prof Josef Bafort, Prinz Leopold Institute, Antwerpen, Belgium [62] and Py was a personal gift from Prof John Playfair, London [63] to GEG. Parasite stability was prepared as previously described [64] and stored in liquid nitrogen.

Three experimental groups of mice were studied: non-infected (*n* = 10), PbA-infected (*n* = 7) and Py-infected (*n* = 8). Infection was induced by intra-peritoneal injection of 1 × 10^6^ infected erythrocytes [39,64]. Mice were euthanised seven days post-inoculation. Parasitaemia was monitored by counting 500 erythrocytes in Diff-Quick-stained thin blood smears.

### 4.2. Blood Sampling and MV Preparation

Mouse venous blood was collected by retro-orbital venepuncture under anaesthesia into 0.129 mol/L sodium citrate (ratio of blood to anticoagulant 4:1). Samples were centrifuged at 1500× *g* for 15 min at room temperature. Harvested supernatant was further centrifuged at 18,000× *g* for 4 min, twice, to achieve platelet-free plasma (PFP) and MV pellets. MV numbers were assessed as previously described [12].

### 4.3. Lipid Extraction

Lipids were extracted from 18k pallets enriched from 1 mL of PFP following the MTBE protocol as described by Matyash et al. [65]. In brief, 300 µL of methanol containing 5 µL Splash Lipidomix deuterated standard (Avanti, USA—purchased from Sigma, Sydney, NSW, Australia) was in Eppendorf tubes cooled on ice. Samples were vortexed briefly and incubated on ice for 10 min. MTBE (1000 µL) was added to the tubes, which then were vortexed, and the contents were allowed to mix on a rotating shaker at 4 °C for 1 h. Optima level H_2_O (250 µL) was added before samples were vortexed briefly and kept on ice for 10 min to allow phase separation. Following this, samples were centrifuged for 10 min at 10,000× *g* in a tabletop centrifuge set to 4 °C and 900 μL of the MTBE/Methanol top phase was transferred to 1.5 mL Eppendorf tubes. Lipid extracts were stored at −80 °C until analysed by liquid chromatography–mass spectrometry (LCMS).

### 4.4. Liquid Chromatography–Mass Spectrometry

For liquid chromatography, 900 µL aliquots of the lipid extracts were dried in a speed vacuum and reconstituted in 100 µL of isopropanol:methanol (2/1 *v*/*v*), vortexed for 20 s twice and centrifuged at 10,000× *g* for 30 s. Lipid extracts were transferred to glass vials with glass inserts and Teflon caps prior to analysis. Reversed-phase, ultra-high performance liquid chromatography (RP-UHPLC) was performed using a Vanquish liquid chromatography (LC) system (Thermo Fisher Scientific, Scoresby, VIC, Australia) fitted with a C30 column (Acclaim 2.1 × 150 mm, 3 μm particle size, Thermo Fisher Scientific, Scoresby, VIC, Australia) held at 10 °C. Two mobile phases were used; A: acetonitrile/water (60/40 *v*/*v*), 10 mM ammonium formate + 0.1% (*v*/*v*) FA and B: isopropanol: acetonitrile (90/10 *v*/*v*), 10 mM ammonium formate + 0.1% (*v*/*v*) FA. For LC-MS operation, 5 µL of the sample was injected onto the column with a solvent flow rate of 400 μL min^−1^. Mass spectrometry (MS) was performed on a Fusion Orbitrap mass spectrometer using targeted and untargeted lipidomic approaches (Thermo Fisher Scientific, Scoresby, VIC, Australia). LipidSearch lipidomic software (Thermo Fisher) was used to annotate and quantify lipid species. The software returns thousands of annotated ions based on the accurate mass, which were manually curated peak by peak. The MS2 data were manually inspected for diagnostic fragments of each lipid species. The retention times of lipids belonging to a class were aligned to further confirm the lipid annotations. Only the lipids that were confirmed by MS2 data and the retention time pattern and were detected from groups were kept for analysis. The data were then exported to Excel, and each lipid class was normalised to one internal standard. The concentrations generated were then sent for statistical analysis.

### 4.5. Nomenclature

The shorthand notation of lipid structures used here is guided by the literature recommendations of Liebisch et al. [66]. Common names of lipid classes were used to ensure easy comprehension of the material. The LIPID MAPS nomenclature (as per the latest update, 2020) of all the lipid classes mentioned in this manuscript can be accessed in the publication by Liebisch et al. [66].

### 4.6. Statistical Analysis

Pre-processing and differential analyses were performed in R using the ‘limma’ package. Raw lipid profiles were log2 transformed and normalised to equalise median absolute values across samples (see Appendix A for pre- vs. post-normalisation profiles). Moderated *t*-test [67] was applied to normalised profiles to rank lipid species in order of evidence for differential expression; *p*-values were adjusted for multiple hypothesis testing using Benjamini and Hochberg correction [68], also known as false discovery rate (FDR) correction. The interquartile range (IQR) is a measure of statistical dispersion defined as the difference between the 75th and 25th percentiles of the data [69]. IQR was used to identify invariant lipids (i.e., IQR ≤ 1). For correlation analysis, invariant lipids were removed (Appendix A); 121 lipid ions out of 302 were retained for subsequent analysis. Pearson correlation was performed to estimate the pairwise correlation among the 12 non-invariant lipids across all samples. The correlation matrix was visualised using the ‘corrplot’ R package, where lipids were ordered using hierarchical clustering with the ‘complete’ agglomeration method.

## Figures and Tables

**Figure 1 ijms-24-00501-f001:**
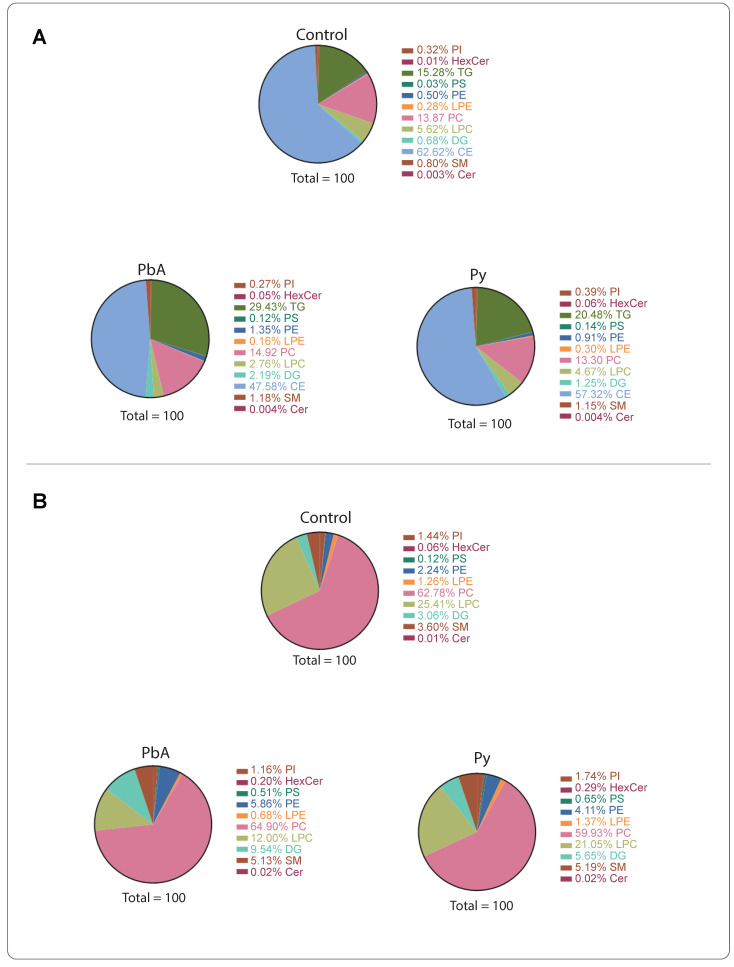
Lipid classes in plasma microvesicles (MV) from uninfected controls, PbA-infected and Py-infected mice. (**A**) All lipid classes. (**B**) Levels without triglycerides (TG) and cholesteryl esters (CE).

**Figure 2 ijms-24-00501-f002:**
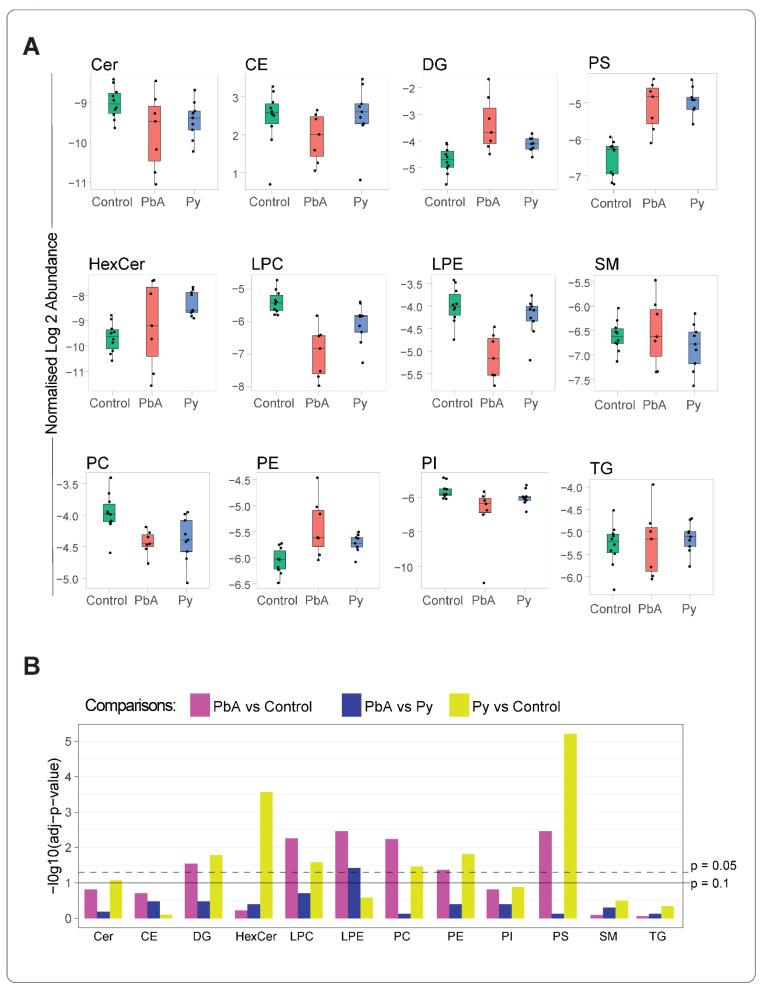
Altered lipid class levels in MV (18k PFP) from the three groups of mice. The mean value of the concentrations (pmol/uL) of lipid ions belonging to a lipid class was estimated. The lipid ion concentration was calculated by normalising to the respective internal standards (**A**) Boxplots comp the distribution statistics (the minimum, maximum, median and the first and third quartiles) of the 12 lipid classes stratified to the groups of control (green), PbA (red) and Py (blue). Each dot corresponds to a sample, with its value representing the mean concentration of lipid ions of the respective lipid class. Data were log-transformed. (**B**) Significance of differences was assessed using Student’s *t*-test followed by FDR correction; adjusted *p*-values of 0.05 and 0.1 are marked with dashed and solid lines, respectively. Abbreviations as in Figure 1.

**Figure 3 ijms-24-00501-f003:**
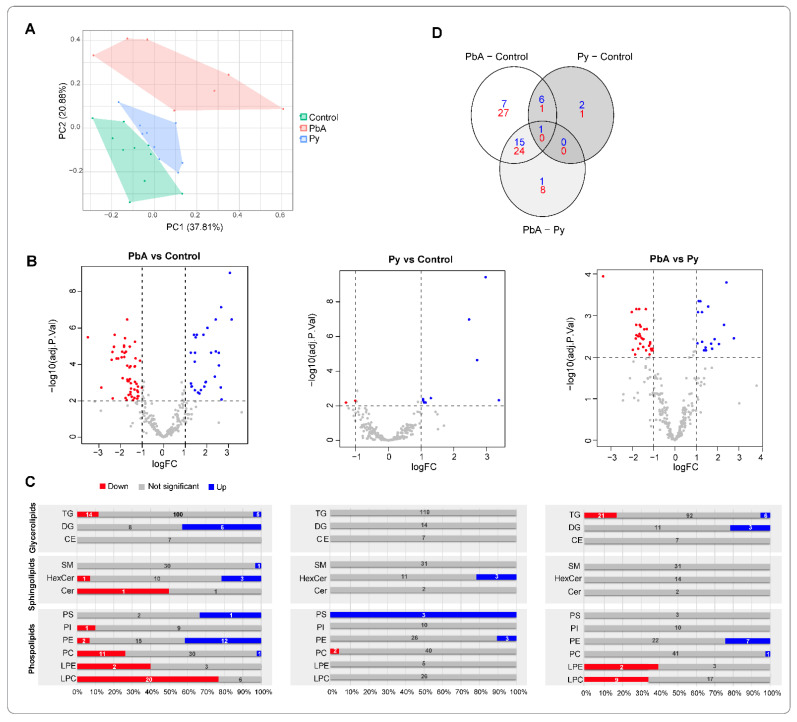
Differentially expressed lipids. (**A**) Principal component analysis (PCA) of identified lipids (pmol/µL). (**B**) Volcano plots showing differentially expressed lipids in the various pairwise comparisons. (**C**) Bar charts representing the proportion of differentially expressed ions across each lipid graph. Blue, grey and red bars show the percentage of elevated, not significant and decreased level of lipid molecules in each type. Numbers on top show the actual number of differentially expressed ions in each type. (**D**) Venn diagram of the distribution of differentially expressed lipids in the various comparisons. Significance is based on two-fold increase or decrease in lipid amounts plus an adjusted *p*-value < 0.01 based on moderated *t*-test and false discovery rate (FDR) correction.

**Figure 4 ijms-24-00501-f004:**
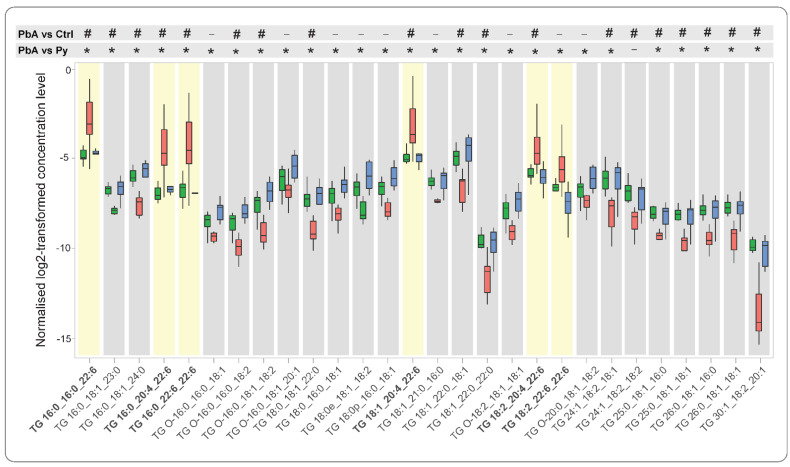
Docosahexaenoic acid (DHA 22:6) containing TG lipids are higher during cerebral malaria (CM). Comparison of levels of characterised TG species from plasma MVs between PbA (red) vs. control (green) and Py (blue). Yellow boxes highlight DHA 22:6 containing TG lipids that are exclusively elevated in PbA vs. control and Py. To display the statistical significance of differences, symbols (-) represents no statistical difference, (#) for PbA vs. control and (*) for PbA vs. Py based on adjusted *p*-value < 0.01. There are no differences in TGs between Py and control.

**Figure 5 ijms-24-00501-f005:**
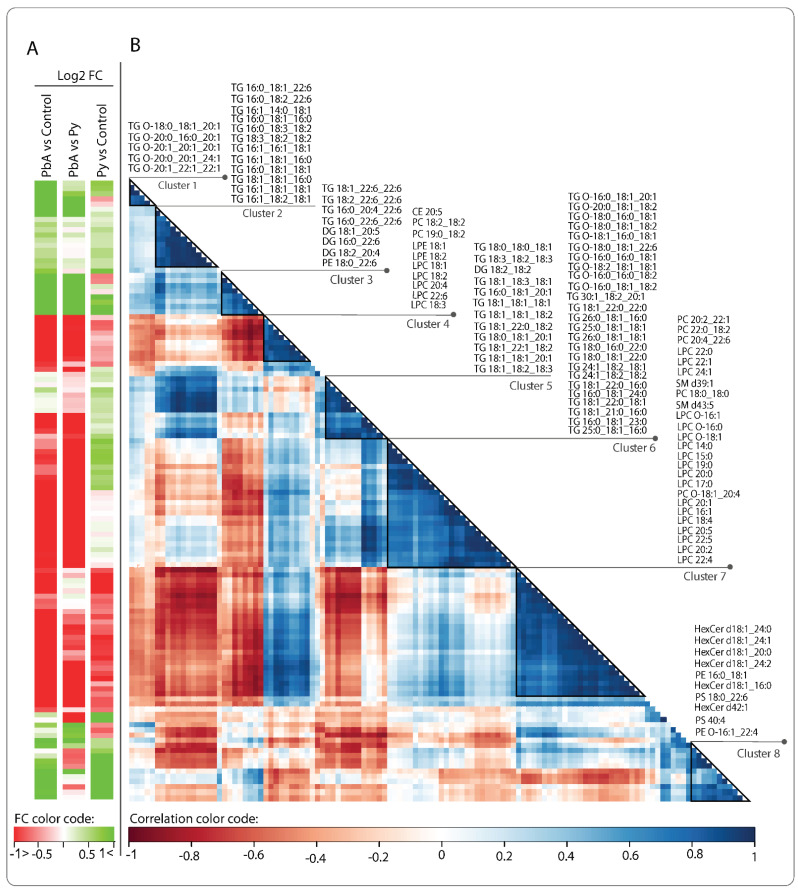
Correlation plot of the 121 retained lipids across all the samples. (**A**) Fold changes (FC) of lipid amounts in the three categories of comparisons. Green denotes an increase and red a decrease. (**B**) Hierarchical clustering of the lipid–lipid correlation matrix, where the pairwise Pearson correlation was conducted across all samples in the three categories. Rows and columns correspond to the 121 retained lipid species. Dark blue triangles indicate clusters (1–8) of strongly positively correlated lipids. Cluster numbers and corresponding lipid names are shown on the right.

**Table 1 ijms-24-00501-t001:** Individual lipid molecules found at significantly different levels in PbA vs. control. Ions elevated (blue) and decreased in PbA are (red) were listed. Log2 fold-change (LogFC) and adjusted *p*-values are listed (i.e., moderated *t*-test and FDR correction). Ions are grouped into neutral lipids, sphingolipids and phospholipids and sorted alphabetically within each group.

Lipid Ion	Log FC	Adjusted *p*-Value	Lipid Ion	Log FC	Adjusted *p*-Value
**Glycerolipids**			**Phospholipids**		
DG 16:0_22:6	3.15	3.45 × 10^−7^	LPC 18:1	−1.58	0.000144
DG 18:1_20:4	2.19	2.22 × 10^−5^	LPC 18:2	−2.11	4.64 × 10^−5^
DG 18:1_20:5	2.37	0.000461	LPC 18:4	−2.28	2.35 × 10^−6^
DG 18:1_22:6	1.72	0.002531	LPC 19:0	−1.75	1.94 × 10^−5^
DG 18:2_20:4	2.41	1.90 × 10^−5^	LPC 20:0	−1.81	2.24 × 10^−5^
DG 18:2_22:6	1.95	0.000892	LPC 20:1	−1.84	9.10 × 10^−6^
TG 16:0_16:0_22:6	1.94	0.000949	LPC 20:2	−2.3	1.06 × 10^−5^
TG 16:0_18:1_23:0	−1.19	0.001628	LPC 20:4	−1.54	0.000123
TG 16:0_18:1_24:0	−1.54	0.003227	LPC 20:5	−2.37	2.04 × 10^−5^
TG 16:0_20:4_22:6	2.56	0.000112	LPC 22:1	−1.23	0.007528
TG 16:0_22:6_22:6	2.64	0.001803	LPC 22:4	−2.41	5.44 × 10^−5^
TG O-16:0_16:0_18:2	−1.2	0.005488	LPC 22:5	−2.18	4.64 × 10^−5^
TG O-16:0_18:1_18:2	−1.36	0.006879	LPC 22:6	−1.33	0.000892
TG 18:0_18:1_22:0	−1.62	0.004668	LPE 18:1	−1.79	3.58 × 10^−6^
TG 18:1_20:4_22:6	1.88	0.001628	LPE 18:2	−1.92	1.94 × 10^−5^
TG 18:1_22:0_18:1	−1.71	0.009094	PC 15:0_18:2	−1.33	5.73 × 10^−6^
TG 18:1_22:0_22:0	−1.9	0.000559	PC 16:0_20:5	−1.01	0.00189
TG 18:2_20:4_22:6	1.59	0.003492	PC O-16:0_20:4	−1.34	0.00013
TG 24:1_18:2_18:1	−1.77	0.006916	PC O-16:0_22:4	−1.23	0.002657
TG 24:1_18:2_18:2	−1.48	0.005911	PC 16:1_20:4	−1.3	4.54 × 10^−5^
TG 25:0_18:1_16:0	−1.26	0.001152	PC 17:0_18:2	−1.1	6.07 × 10^−5^
TG 25:0_18:1_18:1	−1.63	0.000827	PC 18:0_22:6	1.28	0.001628
TG 26:0_18:1_16:0	−1.52	0.001061	PC O-18:1_20:4	−1.57	1.04 × 10^−5^
TG 26:0_18:1_18:1	−1.67	0.000815	PC 18:2_18:2	−1.95	3.99 × 10^−5^
TG 30:1_18:2_20:1	−3.51	3.19 × 10^−6^	PC 19:0_18:2	−1.69	3.45 × 10^−7^
**Sphingolipids**			PC 20:4_22:6	−1.17	0.0055
SM 36:0;O2	1.65	0.003903	PC 22:0_18:2	−1.57	0.000624
HexCer 18:1;O2/16:0	3.06	9.36 × 10^−10^	PE 16:0_18:1	2.03	9.88 × 10^−7^
HexCer 41:1;O2	−2.36	0.007303	PE 16:0_18:2	1.4	2.35 × 10^−6^
HexCer 42:1;O2	2.67	0.00836	PE 16:0_20:4	1.53	2.35 × 10^−6^
			PE 16:0_20:5	1.44	0.002587
**Phospholipids**			PE 18:0_20:5	1.44	6.91 × 10^−5^
LPC 14:0	−1.69	0.000114	PE 18:0_22:6	2.66	7.25 × 10^−8^
LPC 15:0	−1.83	3.83 × 10^−5^	PE O-18:1_18:2	−1.36	0.002351
LPC O-16:0	−1.19	0.003254	PE 18:1_18:1	1.3	0.001572
LPC O-16:1	−1.54	0.004291	PE 18:1_20:4	1.25	2.22 × 10^−5^
LPC 16:1	−1.84	1.04 × 10^−5^	PE 18:1_22:6	1.48	3.26 × 10^−6^
LPC 17:0	−1.34	0.000123	PI O-34:3	−2.89	0.001838
LPC O-18:1	−1.68	0.000668	PS 18:0_22:6	2.4	3.45 × 10^−7^

**Table 2 ijms-24-00501-t002:** Individual lipid molecules found at significantly different levels in Py vs. control. Ions elevated (dropped) in Py are coloured in blue (red). Log2 fold-change and adjusted *p*-values are listed (i.e., moderated *t*-test and FDR correction). Ions are grouped into sphingolipids and phospholipids and sorted alphabetically within each group.

Lipid Ion	Log FC	Adjusted *p*-Value
**Sphingolipids**		
HexCer 18:1:O2/16:0	2.97	3.76 × 10^−10^
HexCer 42:1;O2	3.38	0.004698
**Phospholipids**		
PC 18:2_22:6	−1.01	0.005038
PC 20:4_22:6	−1.3	0.006469
PE 16:0_18:1	1.06	0.005038
PE O-16:1_22:4	1.3	0.003583
PE 18:0_22:6	1.09	0.006469
PS 18:0_20:3	1.14	0.006469
PS 18:0_22:6	2.46	1.05 × 10^−7^
PS 40:4	1.06	0.004093

**Table 3 ijms-24-00501-t003:** Individual lipid molecules found at significantly different levels in PbA vs. Py. Ions elevated (dropped) in PbA are coloured in blue (red). Log2 fold-change and adjusted *p*-values are listed (i.e., moderated *t*-test and FDR correction). Ions are grouped into neutral lipids and phospholipids (located in the tables in a vertical fashion near each group of lipids) and sorted alphabetically within each group.

Lipid Ion	Log FC	Adjusted *p*-Value	Lipid Ion	Log FC	Adjusted *p*-Value
**Glycerolipids**			**Phospholipids**		
DG 16:0_22:6	2.41	0.00016	LPC 16:1	−1.16	0.00438
DG 18:1_20:4	1.37	0.006795	LPC 18:1	−1.21	0.005347
DG 18:2_20:4	1.46	0.006804	LPC 18:2	−1.62	0.003162
TG 16:0_16:0_22:6	1.86	0.003682	LPC 18:4	−1.43	0.002132
TG 16:0_18:1_23:0	−1.16	0.004729	LPC 20:1	−1.05	0.006804
TG 16:0_18:1_24:0	−1.63	0.004729	LPC 20:2	−1.5	0.003682
TG 16:0_20:4_22:6	2.3	0.001677	LPC 20:4	−1.1	0.006804
TG 16:0_22:6_22:6	2.77	0.003539	LPC 22:4	−1.83	0.003564
TG O-16:0_16:0_18:1	−1.32	0.002115	LPC 22:5	−1.82	0.001677
TG O-16:0_16:0_18:2	−1.8	0.000701	LPE 18:1	−1.36	0.000701
TG O-16:0_18:1_18:2	−2.02	0.000832	LPE 18:2	−1.68	0.000701
TG O-16:0_18:1_20:1	−1.35	0.006804	PC 18:0_22:6	1.27	0.004289
TG 18:0_18:1_22:0	−1.72	0.006201	PE 16:0_18:2	1.11	0.000449
TG O-16:0_16:0_18:1	−1.6	0.001852	PE 16:0_20:4	1.08	0.000832
TG O-16:0_18:1_18:2	−1.72	0.003162	PE 16:0_20:5	1.44	0.005871
TG O-18:1_16:0_18:1	−1.65	0.001677	PE 16:0_22:6	1.27	0.000832
TG 18:1_20:4_22:6	1.74	0.006201	PE 18:0_20:5	1.06	0.004658
TG 18:1_21:0_16:0	−1.05	0.006201	PE 18:0_22:6	1.56	0.000608
TG 18:1_22:0_18:1	−1.97	0.006618	PE 18:1_22:6	1.2	0.000455
TG 18:1_22:0_22:0	−1.8	0.002948			
TG O-18:2_18:1_18:1	−1.48	0.005676			
TG 18:2_20:4_22:6	1.7	0.004729			
TG 18:2_22:6_22:6	2.06	0.004839			
TG O-20:0_18:1_18:2	−1.19	0.008578			
TG 24:1_18:2_18:1	−1.87	0.008578			
TG 25:0_18:1_16:0	−1.12	0.006236			
TG 25:0_18:1_18:1	−1.56	0.003502			
TG 26:0_18:1_16:0	−1.47	0.003682			
TG 26:0_18:1_18:1	−1.66	0.002776			
TG 30:1_18:2_20:1	−3.36	0.000114			

## Data Availability

The lipid profiles and R codes used to generate the results presented in this study are available at https://github.com/VafaeeLab/CM_Lipidomics.

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
