# Peer review of "Investigation of Plasma-Derived Lipidome Profiles in Experimental Cerebral Malaria in a Mouse Model Study"

_ijms, 2022, doi:10.3390/ijms24010501_

Round 1
Reviewer 1 Report
In this study, the authors revealed the difference in the lipidomic profile of a defined plasma fraction between the test groups of experimental Malaria using an untargeted lipidomic approach. It discloses some interesting results for further exploring the pathogenetic mechanisms. However, several points should be addressed before the study being considered for publishing.
1. Plasma particle fractions and their physiological roles should be briefly introduced. Which particle fraction is the studied plasma microvesicles belonging to? In addition, it is confused that “Lipids were extracted from 1 mL of PFP following …” was mentioned in 4.3. Lipid extraction, but the lipid analytical results of MV pellets were described through the text. The role of lipids in the pathophysiology of malaria infection should be emphasized more in the introduction.
2. In section 4.4, the data processing should be described as a supplement. It should also be explained how to did the identifying of lipid classes and molecular species. In Figure 2, the unit of lipid content was indicated by “(pmol/ul)”, is this concentration normalized to the internal standard? Did you sure that the amount of MV pellets being the same in each sample for the lipid extraction? Why not do the normalization by the amount of MV pellets or the signal sum of the total detected lipids?
3. In 4.5. Nomenclature, please update the last recommendations from LIPID MAPS, and adopt it in the text of this study.
4. In Figure 2, the mention in figure legend “the groups of control (green) …” was different as shown in the content of figure. The number of black dots (I guessed that they indicate each identified lipid species) should be sample sizes but not lipid species number * sample sizes.
5. In Table 1, what is “og FC”?, where is the “adjusted p-values”? and what kind of lipid ion was listed in the table, are they ions? The group name “glycerolipids” may be more suitable than that of “neutral lipids”. In all tables, the arrangement of contents should be improved to be more apparent to the eye.
6. The discussion should be emphasized more in explaining the relationship between the changed lipids and the specific pathological symptoms of CM differing from that infected with P. yoelii.
Author Response
We thank the IJMS Editor and respected reviewers for providing us with the opportunity to resubmit our manuscript with revisions. In particular, we are grateful for the time the reviewers gave to provide us with constructive feedback. We addressed reviewers’ comments as detailed in the point-by-point responses below. Changes in the Manuscript are highlighted.
Reviewer #1:
In this study, the authors revealed the difference in the lipidomic profile of a defined plasma fraction between the test groups of experimental malaria using an untargeted lipidomic approach. It discloses some interesting results for further exploring the pathogenetic mechanisms. However, several points should be addressed before the study being considered for publishing.
We thank the reviewer for endorsing our manuscript. The comments were carefully addressed as below:
- Plasma particle fractions and their physiological roles should be briefly introduced. Which particle fraction is the studied plasma microvesicles belonging to? (Lipid fraction size or all particle in plasmae??) In addition, it is confused that “Lipids were extracted from 1 mL of PFP following …” was mentioned in 4.3. Lipid extraction, but the lipid analytical results of MV pellets were described through the text.
Response: We have now added a brief description in the introduction (page 4, lines 56 to 74, Ref # 28-38) and revised the method (page 12, lines 349) to address the comment. Briefly, we used two-step centrifugation of 1500 g and 1800 g to have platelet-free plasma and assess the lipid profile. We believe, based on our previous study, the majority of fractions in the 18 k pallet contain large EVs or MVs.
The role of lipids in the pathophysiology of malaria infection should be emphasized more in the introduction.
Response: In the revised version, we further discussed in the Introduction and Discussion the role of lipids and fatty acids in malaria (page 3, lines 56 – 59) and (page 9, lines 235 – 242 and page 10, lines 263 – 272, Ref# 27, 54-55,57).
- In section 4.4, the data processing should be described as a supplement. It should also be explained how to did the identifying of lipid classes and molecular species
Response: We enhanced data processing explanation and included more information on lipid identification and explained the process in detail (page 13, lines 377-383)
In Figure 2, the unit of lipid content was indicated by “(pmol/ul)”, is this concentration normalized to the internal standard? Did you sure that the amount of MV pellets being the same in each sample for the lipid extraction? Why not do the normalization by the amount of MV pellets or the signal sum of the total detected lipids?
Response: The concentration is normalised to the internal standard and the same volume of plasma. We have now included this information in Figure 2 legend (page 19, lines 9-10).
The amount of MV pellet in each sample cannot be measured accurately as the pellet is not dried at the end of the enrichment process. Therefore, we surmised that normalisation to the weight of MV pellet would not be accurate. We did not weigh the MV pellets. Thus, it is not possible to normalise to the wet weight at this stage. However, the same volume of plasma was used to enrich MVs. The sample processing workflow was rigorously quality-controlled to ensure consistency in MV enrichment and lipid extraction.
The instrument’s response for different lipid species, even within the same lipid class, can differ significantly depending on numerous factors. For example, the instrument’s response to phospholipids is dependent on acyl chain length, acyl chain unsaturation, the structure of the polar head group, total lipid concentration, solvent composition, and instrument settings. Therefore, normalising to the signal sum of total detected lipids is not accurate.
However, normalising to the signal sum of lipids per class is possible. Normalisation to the sum concentration of total lipids (calculated using the respective internal standards) is also possible.
The “minimum standards” established by the Lipidomics Standards Initiative require that at least one internal standard per class should be used for lipid quantification. Therefore, we followed the guidelines set by the Lipidomics Standards Initiative to ensure community-wide consistency in lipidomic workflows.
- In 4.5. Nomenclature, please update the last recommendations from LIPID MAPS, and adopt it in the text of this study.
Response: To address this comment, we checked the annotations in the manuscript against the latest update of LIPID MAPS and highlighted some of them in the manuscript as an example and revised the method section and updated all the figures and tables accordingly (page 14, lines 387 -391, Ref #66, Figures 1 to 5, Tables 1 to 3).
- In Figure 2, the mention in figure legend “the groups of control (green) …” was different as shown in the content of figure (the figure legend was revised control as red and PbA as green and the results related to fig 2 were revised in the text. The number of black dots (I guessed that they indicate each identified lipid species) should be sample sizes but not lipid species number * sample sizes.
Response: Thanks for noting that! To comprehensively address this comment, we:
- Corrected the caption of Figure 2 (page 20, line 8-17) and made the colour code of the three groups consistent across all figures (referring to the comment by Reviewer 2).
- Modified the boxplots to compare the mean concentration of lipids belonging to the same group within each sample and updated the figure accordingly such that each black dot represents a sample, with its value corresponding to the mean concentration of the lipid ions belonging to the respective class.
- Included Figure 2B to show the adjusted p-values (t-test followed by FDR correction) comparing the significance of differences between any two groups
- Revised the related paragraph in the results (page 5, lines 101-109) and discussion (page 8, lines 218-220 and page 9, lines 246-249) as well as the figure caption (page 20, lines 8-17)and Table legends ( page 21) to precisely explain the figure and findings.
- In Table 1, what is “log FC”?, where is the “adjusted p-values”? and what kind of lipid ion was listed in the table, are they ions? The group name “glycerolipids” may be more suitable than that of “neutral lipids”. In all tables, the arrangement of contents should be improved to be more apparent to the eye.
Response: To address this comment, we revised the Tables and the headings to improve the reliability and clarity of the content (page 23-25). The words “Neutral lipids” were replaced with “glycerolipids” in all Tabels and Figure 3. The LogFC stands for “log2 fold change”. The “adjusted p-value” is the p-value adjusted by false discovery rate correction as described in the Methods (Page 14, lines 400-406, Ref# 68-69).
- The discussion should be emphasized more in explaining the relationship between the changed lipids and the specific pathological symptoms of CM differing from that infected with P. yoelii.
Response: The discussion has been revised to address the comment (page 9) as follow:
- We added a sentence to describe the Malaria symptoms in CM and non-CM mice (page 8, lines 211-216, Ref# 19, 48-49).
- We added a sentence about the effect of PLA , LPE and LPC on immunity responses in the host (page 9, lines 235-241, Ref # 54-55)
- A new paragraph has been added about the lipid species and their different levels in PbA and Py infected mice and their possible role in the infection (page 10, lines 253-272)
Reviewer #2:
In this manuscript, Batarseh at al., quantify and analyse the lipid profiles of plasma microvessicles in control mice or mice infected with P. yoelii (no cerebral malaria) and P. berguei (CM). After recent evidence of the role of microvesicles and lipidomics in vascular disease pathogenesis and malaria, the paper is timely and overall, well written. However, there are a few aspects of the analysis and interpretation of the results that need to be addressed.
We thank the reviewer for acknowledging the value of our manuscript. We carefully address all the comments as follows:
MAJOR:
The paper presents consists of 5 figures and 3 tables that analyze the same dataset. The first two figures quantify the lipid classes, and the remaining figures and tables focus on the analysis and quantification of individual lipids. The major conclusion and interpretation of the authors is that the experimental cerebral malaria model (P. berguei) presents a unique lipid profile compared to P. yoelii infected mice and control mice, based on the analysis on individual lipids. However, significant differences arise between P. yoelii and P. berguei infection when the lipid class is analyzed (Figure 2). Since the paper does not show any experimental and mechanistic data on how differences in lipid composition could cause vascular dysfunction in experimental CM, what is the biological reason to draw conclusions from individual lipid class composition and not lipid classes? I would suggest that authors comment on this in the discussion
Response: We aimed to explore if lipid species are changing at the lipid species level instead of class level, as individual lipids can be interesting for directing new research efforts. A sentence added to the conclusion to address the comment (page 11, lines 307 to 311). Same as the response to previous discussion comment (we enhanced the discussion and added some sentences supporting from the literature
- In some figures the authors explicitly mention that they have adjusted for multiple comparisons by false discovery rate, but not in others (Figure 2). Please, specify if all figures present adjusted p-values by FDR or provide the adjusted values.
Response: Thanks for noting that! Figure 2 was the only figure where non-adjusted p-values were reported. In the revised manuscript, we modified Figure 2 to show the adjusted p-values (Figure 2B). We also modified the relevant sections in the manuscript, i.e., Figure’s caption(page 20, lines 7-17), Results (page 5, lines 101-109), and Discussions (page 8, lines 218-220 and page 9, lines 246-249) (see also the response to comment #4 of Reviewer #1)
- Does the hierarchical clustering of the lipid-lipid correlation matrix show the comparison of the three groups? Or just a pairwise comparison? If so, which groups. I would suggest to extend the description of how the analysis was done in the result section, figure legend and methods.
Response: The correlation matrix represents pairwise Pearson correlations among 121 non-invariant lipids across all samples. Therefore, hierarchical clustering identifies groups of lipid species whose concentrations move in tandem across all samples. However, we also annotated the lipid species with their log fold-change between the groups (as vertical annotation bars adjacent to the correlation matrix). As suggested by the reviewer, we clarified the explanation in the manuscript (page 13: lines 399-406, Ref# 68-69, and Figure Legends 5: page 21, lines 41 – 43).
Association vs causality: The manuscript does not show any mechanistic experiment showing that the changes in microvesicle’s lipid composition cause pathogenesis. Although in most sections authors do not assume causality, still some sentences still imply it. I would suggest to rephrase them. Examples:
- These results suggest that experimental CM is characterised by specific changes in lipid composition of circulating MV, pointing towards triglycerides (TG), especially docosahexaenoic acid (DHA 22:6) containing species, phosphatidylethanolamine (PE), LPC, LPE, and diacylglycerol (DG) as potentially important players in CM pathogenesis
Response: The Abstract has been rephrased to address the comment. We have added a sentence about the observed changes in lipid species and rewrote the concluision (page 2, lines 11 to 20).
- We also found that PE levels were significantly higher in PbA mice compared to those from uninfected control and Py. On the other hand, total PS levels were significantly higher in both PbA and Py compared to uninfected control, suggesting a role for PS in malarial infection but not CM pathogenesis specifically.
Response: To address this comment, we changed the above sentences based on the updated figure 2 as well as any other occasions which may infer causality throughout the manuscript (page 9, lines 246 to 248).
Description of figures in results: In 4 in which results of Figure 2 are introduced, the description of the text does not correspond with the significance showed in the figure. Some of the lipid classes described to be significant are not in the figure, and others with significance in the figure are missing.
Response: We modified the Figure to show adjusted p-values of all pairwise comparisons (Figure 2B). We also carefully reviewed and modified results to ensure that the descriptions in the text were consistent with the figure. Please note that in the former version of the manuscript, significance analyses in Figure 2 were based on the non-adjusted p-values, which were adjusted during the revision. Therefore, some of the observations, while consistent overall, were changed, which have been reflected in the revised manuscript (page 5, lines: 101 to 109).
Color scheme in figures: The authors assign different colors to each experimental group. Example: Control red in fig 2, green in fig 3 and 4. I would suggest using the same color for each experimental group in all figures.
Response: Thanks for noting that! We made the colour choices of each group consistent across all figures (c.f., Figures 2, 3, and 4). We also updated the lipid annotations in all figures and tables and make them more readble (page 23-25)
Discussion: the authors discuss a reported increase in LPC in platelets of myocardial infection and in schistosomiasis and then suggest that the decreased LPC level in P. berguei infection in preventing tissue repair. The balance of inflammation and repair is quite complex and might be difficult to understand for many readers. I would suggest rephrasing this paragraph to improve clarity.
Response: We thank the reviewer for this recommendation. We removed this sentence and revised the discussion to improve the content, add more details and enhance the clarity to address this comment. (see also the response to comment #6 of Reviewer #1)
Minor:
Abstract: The conclusion sentence in the abstract enumerates multiple lipid classes “triglycerides (TG), especially docosahexaenoic acid (DHA 22:6) containing species, phosphatidylethanolamine (PE), LPC, LPE, and diacylglycerol (DG)” without mentioning them before on the text. This comes out of the blue. Please, either mention before how they are associated with experimental CM or finish with a more general statement.
Response: We rewrote the final part of the sentence to make it more general and conclusive (page 2, lines 16-19).
Introduction:
“The murine CM model has limits [13] but also several positive aspects [14- 19], which makes it a valuable tool [20].” I would suggest to briefly mention the limitations.”
Response: To address the comment, we have added the limitations and positive aspects of the murine CM model to the paragraph (page 3, lines 38-45, Ref # 13-19).
Results:
“Compared to those from controls, MV from PbA-infected mice showed a doubling of their proportion of triglycerides (TG), a 25% reduction (47.6 versus 62.6%) in their cholesteryl ester (ChE) proportion, and a 50% reduction (2.8 versus 5.6%) in their lysophosphatidylcholine (LPC) content (Figure 1A)”. For consistency, either at the percentage in both groups for TG or remove the percentages for ChE and LPC.
Response: We now added the percentage of TG “(29.43% vs 15.28%)” to the sentence (page 4, line85).
Figure 3 legend: The abbreviation for differentially expressed (DE) is used and not used elsewhere in the manuscript. The “DE” was removed from the sentence.
Response: The abbreviation “DE” was removed from the sentence (page 20, line24). We also updated the figure based on the latest LIPIDMAP recommendation.
Table 1, 2 and 3 legend: The parenthesis that shows that decreased lipids are in blue in parenthesis is confusing.
Response: We do apologise for the confusion. All the tables have been revised to be more appealing (page 21 line 54 and page 23-25).
9: Describe IQR (interquartile range?)
Response: The words “interquartile range” were added to the sentence (page 7, line174).

Reviewer 2 Report
In this manuscript, Batarseh at al., quantify and analyze the lipid profiles of plasma microvessicles in control mice or mice infected with P. yoelii (no cerebral malaria) and P. berguei (CM). After recent evidences of the role of microvesicles and lipidomics in vascular disease pathogenesis and malaria, the paper is timely and overall well written. However, there are a few aspects of the analysis and interpretation of the results that need to be addressed.
MAJOR:
The paper presents consists of 5 figures and 3 tables that analyze the same dataset. The first two figures quantify the lipid classes, and the remaining figures and tables focus on the analysis and quantification of individual lipids. The major conclusion and interpretation of the authors is that the experimental cerebral malaria model (P. berguei) presents a unique lipid profile compared to P. yoelii infected mice and control mice, based on the analysis on individual lipids. However, significant differences arise between P. yoelii and P. berguei infection when the lipid class is analyzed (Figure 2). Since the paper does not show any experimental and mechanistic data on how differences in lipid composition could cause vascular dysfunction in experimental CM, what is the biological reason to draw conclusions from individual lipid class composition and not lipid classes? I would suggest that authors comment on this in the discussion
Statistical analysis:
- In some figures the authors explicitly mention that they have adjusted for multiple comparisons by false discovery rate, but not in others (Figure 2). Please, specify if all figures present adjusted p-values by FDR or provide the adjusted values.
- Does the hierarchical clustering of the lipid-lipid correlation matrix show the comparison of the three groups? Or just a pairwise comparison? If so, which groups. I would suggest to extend the description of how the analysis was done in the result section, figure legend and methods.
Association vs causality: The manuscript does not show any mechanistic experiment showing that the changes in microvesicle’s lipid composition cause pathogenesis. Although in most sections authors do not assume causality, still some sentences still imply it. I would suggest to rephrase them. Examples:
- These results suggest that experimental CM is characterised by specific changes in lipid composition of circulating MV, pointing towards triglycerides (TG), especially docosahexaenoic acid (DHA 22:6) containing species, phosphatidylethanolamine (PE), LPC, LPE, and diacylglycerol (DG) as potentially important players in CM pathogenesis
- We also found that PE levels were significantly higher in PbA mice compared to those from uninfected control and Py. On the other hand, total PS levels were significantly higher in both PbA and Py compared to uninfected control, suggesting a role for PS in malarial infection but not CM pathogenesis specifically.
Description of figures in results: In page 4 in which results of Figure 2 are introduced, the description of the text does not correspond with the significance showed in the figure. Some of the lipid classes described to be significant are not in the figure, and others with significance in the figure are missing.
Color scheme in figures: The authors assign different colors to each experimental group. Example: Control red in fig 2, green in fig 3 and 4. I would suggest using the same color for each experimental group in all figures.
Discussion: the authors discuss a reported increase in LPC in platelets of myocardial infection and in schistosomiasis and then suggest that the decreased LPC level in P. berguei infection in preventing tissue repair. The balance of inflammation and repair is quite complex and might be difficult to understand for many readers. I would suggest rephrasing this paragraph to improve clarity.
Minor:
Abstract: The conclusion sentence in the abstract enumerates multiple lipid classes “triglycerides (TG), especially docosahexaenoic acid (DHA 22:6) containing species, phosphatidylethanolamine (PE), LPC, LPE, and diacylglycerol (DG)” without mentioning them before on the text. This comes out of the blue. Please, either mention before how they are associated with experimental CM or finish with a more general statement.
Introduction:
“The murine CM model has limits [13] but also several positive aspects [14- 19], which makes it a valuable tool [20].” I would suggest to briefly mention the limitations.”
Results:
“Compared to those from controls, MV from PbA-infected mice showed a doubling of their proportion of triglycerides (TG), a 25% reduction (47.6 versus 62.6%) in their cholesteryl ester (ChE) proportion, and a 50% reduction (2.8 versus 5.6%) in their lysophosphatidylcholine (LPC) content (Figure 1A)”. For consistency, either at the percentage in both groups for TG or remove the percentages for ChE and LPC.
Figure 3 legend: The abbreviation for differentially expressed (DE) is used and not used elsewhere in the manuscript.
Table 1, 2 and 3 legend: The parenthesis that shows that decreased lipids are in blue in parenthesis is confusing.
Page 9: Describe IQR (interquartile range?)
Author Response

(The authors gave the same response as above.)

Round 2
Reviewer 1 Report
Some shorthand notation of lipids have still not fitted the recommendations of ref. 66 in the manuscript, such as HexCer d18:1_16:0 in page 8 and Table 1. HexCer 18:1;OH/16:0 is right. In Tables, sphingomyelin is also one of sphingolipids. Please check again carefully!
Author Response
We thank the IJMS Editor and respected reviewers for providing us with the opportunity to resubmit our manuscript with minor revisions. We are grateful for the time the reviewers gave to provide us with constructive feedback. We addressed the Reviewer 1 comments as detailed below. Changes in the Manuscript are highlighted.
Reviewer #1:
Some shorthand notations of lipids have still not fitted the recommendations of ref. 66 in the manuscript, such as HexCer d18:1_16:0 in page 8 and Table 1. HexCer 18:1;OH/16:0 is right. In Tables, sphingomyelin is also one of sphingolipids. Please check again carefully!
Response: Thanks for the comment. We do acknowledge the need for the harmonisation of lipid annotations and reporting. We carefully reviewed the lipid IDs to match reference 66 and in consultation with a lipid expert at BCAL Diagnostics. To the best of our knowledge, the current annotations are up to date. Should the Reviewer have any concern on a particular annotation, we would thank the Reviewer to kindly specify the suggested changes we need to amend. The changes include:
- Adjusting the related notations in the Results section with the updated lipid notations (page 6, lines 153 to 156).
- Updating Table 1 and 2 with the latest lipid notations, removed sphingomyelin row and moved SM 36:0;O2 to sphingolipid class (pages 23-24).
- Figure 5 has been updated with the latest lipid annotations.
